# Rapid pathogen identification and antimicrobial susceptibility testing in *in vitro* endophthalmitis with matrix assisted laser desorption-ionization Time-of-Flight Mass Spectrometry and VITEK 2 without prior culture

**Lindsay Y. Chun**[1], **Laura Dolle-Molle**[2], **Cindy Bethel**[2], **Rose C. Dimitroyannis**[1], **Blake L. Williams**[1], **Sidney A. Schechet**[1], **Seenu M. Hariprasad**[1], **Dominique Missiakas**[3], **Olaf Schneewind**[3], **Kathleen G. Beavis**[2,4], **Dimitra Skondra**[1] *

**1** Department of Ophthalmology and Visual Science, The University of Chicago Hospitals and Health System, Chicago, Illinois, United States of America, **2** Clinical Microbiology Laboratory, The University of Chicago Hospitals and Health System, Chicago, Illinois, United States of America, **3** Department of Microbiology, The University of Chicago, Chicago, Illinois, United States of America, **4** Department of Pathology, The University of Chicago Hospitals and Health System, Chicago, Illinois, United States of America

* dskondra@bsd.uchicago.edu

## Abstract

### Purpose

Prompt clinical diagnosis and initiation of treatment are critical in the management of infectious endophthalmitis. Current methods used to identify causative agents of infectious endophthalmitis are mostly inefficient, owing to suboptimal sensitivity, length, and cost. Matrix Assisted Laser Desorption-Ionization Time-of-Flight Mass Spectrometry (MALDI-TOF MS) can be used to rapidly identity pathogens without a need for culture. Similarly, automated antimicrobial susceptibility test systems (AST, VITEK 2) provide accurate antimicrobial susceptibility profiles. In this proof-of-concept study, we apply these technologies for the direct identification and characterization of pathogens in vitreous samples, without culture, as an *in vitro* model of infectious endophthalmitis.

### Methods

Vitreous humor aspirated from freshly enucleated porcine eyes was inoculated with different inocula of *Staphylococcus aureus* (*S. aureus*) and incubated at 37°C. Vitreous endophthalmitis samples were centrifuged and pellets were directly analyzed with MALDI-TOF MS and VITEK 2 without prior culture. *S. aureus* colonies that were conventionally grown on culture medium were used as control samples. Time-to-identification, minimum concentration of bacteria required for identification, and accuracy of results compared to standard methods were determined.

**Data Availability Statement:** All relevant data are within the manuscript.

**Funding:** LC, No grant number, Illinois Society for the Prevention of Blindness, http://www. eyehealthillinois.org/, The funders had no role in study design, data collection and analysis, decision to publish, or preparation of the manuscript. LC, No grant number, University of Chicago Pritzker School of Medicine Calvin Fentress Scholarship, https://pritzker.uchicago.edu/scholarly-opportunities, The funders had no role in study design, data collection and analysis, decision to publish, or preparation of the manuscript.

**Competing interests:** The authors have declared that no competing interests exist.

## Results

MALDI-TOF MS achieved accurate pathogen identification from direct analysis of intraocular samples with confidence values of up to 99.9%. Time from sample processing to pathogen identification was <30 minutes. The minimum number of bacteria needed for positive identification was $7.889 \times 10^6$ colony forming units (cfu/µl). Direct analysis of intraocular samples with VITEK 2 gave AST profiles that were up to 94.4% identical to the positive control *S. aureus* analyzed per standard protocol.

## Conclusion

Our findings demonstrate that the direct analysis of vitreous samples with MALDI-TOF MS and VITEK 2 without prior culture could serve as new, improved methods for rapid, accurate pathogen identification and targeted treatment design in infectious endophthalmitis. *In vivo* models and standardized comparisons against other microbiological methods are needed to determine the value of direct analysis of intraocular samples from infectious endophthalmitis with MALDI-TOF MS and VITEK 2.

## Introduction

Infectious endophthalmitis is a serious intraocular infection that can lead to rapid irreversible visual loss despite aggressive medical and surgical management. It is caused by the replication of bacterial, mycobacterial, or fungal agents in the intraocular chambers, and can occur in the setting of surgery, intraocular injections, trauma, contiguous spread from adjacent structures, and endogenous spread through bloodstream sources[1,2]. Prompt clinical diagnosis and initiation of treatment are critical to preserve visual function, as the incidence of infectious endophthalmitis leading to no light perception vision in the affected eye ranges from 23% to 35%[3].

The clinical diagnosis of endophthalmitis incorporates microbiological techniques to determine the causative pathogen to optimize treatment strategy. However, traditional microbiological methods currently used are suboptimal. Intraocular fluids are typically sampled with needle aspiration upon presentation, but the aspiration of vitreous humor can sometimes fail (resulting in a "dry tap"). Cultures of intraocular samples frequently do not reveal any pathogen or can take multiple days to grow on various selective media. Gram stains of intraocular samples are negative in 50–60% of cases, while cultures from vitreous aspirates are negative in up to 55% of cases, and cultures from aqueous aspirates are negative in up to 60% of cases [4–7]. Reported rates for positive cultures that result from vitrectomy samples ranges widely, from 44.6 to 90% of cases, but this range is further complicated by the fact that broad-spectrum intravitreal antibiotics are commonly administered prior to vitrectomy and can prevent the successful identification of intraocular pathogens[4,6].

Techniques that utilize molecular genetics, such as polymerase chain reaction (PCR), sequencing, and microarrays, can identify organisms with accuracy. However, these methods are costly, labor-intensive, require numerous reagents, and are dependent upon the proper selection of complementary primers. Many PCR techniques are unable to provide any information on antimicrobial susceptibilities, and organisms that are difficult to isolate and grow, may require analysis in specialized centers [8]. The emergence of antimicrobial resistance also has serious implications on patient outcome from an epidemiological and public health

perspective, and the accurate and the rapid identification of causative pathogens and the determination of their susceptibility profile can aid in the appropriate management of endophthalmitis[9–11].

Matrix Assisted Laser Desorption/Ionization Time-of-Flight Mass Spectrometry (MALDI-TOF MS) is a promising analytical tool for the expedient identification of pathogens. In the face of increasing antibiotic resistance to empiric antibiotics, the accurate identification of pathogens is important to design targeted treatment and minimize the use of broad-spectrum antibiotics[9]. MALDI-TOF MS laser ionizes whole cell extracts from colonies grown in culture to produce a peptide fingerprint profile, and compares the profile against a proteomic database to identify a pathogen to a species level[3,12–14]. However, MALDI-TOF MS is currently used only in cases where there is already positive organism growth from cultures, and limited data exist for the direct analysis of patient samples[9,13–15].

VITEK 2 is a commercially available antimicrobial susceptibility test (AST) system that utilizes fluorescence-based technology to analyze Gram-positive and Gram-negative bacteria [16,17]. Like MALDI-TOF MS, VITEK 2 is used to analyze microorganisms that are grown in cultures, as opposed to direct analysis of samples, and the AST system is used to analyze isolates that have been already been successfully identified. VITEK 2 AST has shown to have a high degree of agreement with standard methods for determining the minimum inhibitory concentration (MIC) of antibiotics, with a gain-of-time of hours to days and improved reproducibility[16–18]. Additionally, VITEK 2 has been implemented with the use of positive cultures in reported cases of endophthalmitis to determine the AST of causative pathogens[19]. To our knowledge, no data currently exist about using VITEK-2 directly on experimental or patient samples without prior positive culture.

In this report, we investigate the ability of MALDI-TOF MS and VITEK 2 to analyze intraocular samples from *in vitro* models of endophthalmitis in order to rapidly identify and establish the antimicrobial susceptibility profile of the organism involved.

## Deisgn and methods

This study was carried out in strict accordance with the recommendations of, and approved by, the Institutional Biosafety Committee of the University of Chicago (Protocol ID: IBC0610) and the Institutional Animal Care and Use Committee of the University of Chicago (Protocol ID: 72609).

### Vitreous humor preparation

Freshly enucleated porcine eyes were obtained within 6 hours of enucleation, and the ocular surface was sterilized by soaking in 5% Betadine solution for 10 seconds and allowed to dry for at least 2 minutes. Vitreous humor (VH) was aspirated with an 18-gauge needle fitted onto a 10ml syringe using sterile technique, and typical volumes obtained were 1.5-2ml per eye. The VH was filtered through a sterile 0.22μm PES membrane (Whatman, Clifton, NJ) and pooled together. The pooled VH was subjected to a second filter sterilization, aliquoted into 1ml Eppendorf tubes, and stored at -80˚C until use.

### Bacterial stock (BS) preparation

Methicillin-resistant *S. aureus* strain USA300 was used in this *in vitro* model of endophthalmitis because *Staphylococci* are one of the common causative organisms in post-cataract endophthalmitis [4,9,20]. *S. aureus* was streaked onto tryptic soy agar (TSA) and agar plates were incubated for 12 hours. An isolated colony of *S. aureus* was picked to inoculate 1.5ml of VH and incubated with shaking at 37˚C for 7 hours to produce bacterial stock (BS).

### *In vitro* endophthalmitis model

The following *in vitro* endophthalmitis model was applied for MALDI-TOF MS analysis (Vitek MS, Version 3.0, bioMérieux), and VITEK 2 automated antimicrobial susceptibility test (AST) system (VITEK 2, Version 7.01, bioMérieux) at the clinical microbiology laboratory at The University of Chicago Medical Center. 60 μl BS were mixed into 540 μl VH to make serial dilution samples of $10^1$ to $10^{10}$. A 30-μl aliquot from each dilution and a negative control of VH were plated on TSA plates for enumeration of bacteria (colony forming units, cfu) in the BS. Plates were incubated at 37°C for 12 hours. The remainder dilution samples (510 μl) were incubated at 37°C for 11 hours so that bacteria could grow in the vitreous humor, producing the *in vitro* model of infectious endophthalmitis.

Following incubation of the serial dilution samples, a 30-μl aliquot of each sample was plated on TSA plates and incubated at 37°C for 12 hours for post-incubation enumeration of bacteria (Tables 1 and 2). The remaining samples (480 μl) were centrifuged at 16,000 *x g* at room temperature for 20 min to collect bacterial cells in pellets. The supernatant was carefully discarded without disturbing the pellets. The bacterial cell pellets were white and opaque with mucoid consistency (Fig 1). This in vitro endophthalmitis model was duplicated for separate analysis with MALDI-TOF MS and VITEK 2.

**Table 1. Results from direct MALDI-TOF MS analysis of *in vitro* endophthalmitis samples.** Concentrations of *S. aureus* in vitreous humor pre-and post-incubation, descriptions of pellets, and MALDI-TOF MS results with confidence values are depicted. Confidence value scores of ≥60% indicate species-level identification.

| | Pre-incubation (cfu/μl) | Post-incubation (cfu/μl) | Pellet appearance | Spot | Identified organism | Confidence value |
|---|---|---|---|---|---|---|
| $10^1$ dilution | $1.178 \times 10^4$ | $1.15 \times 10^5$ | White, opaque | 1A | *S. aureus* | 99.9% |
| | | | | 1B | *S. aureus* | 99.9% |
| $10^2$ dilution | $1.178 \times 10^3$ | $7.335 \times 10^5$ | White, opaque | 2A | *S. aureus* | 99.9% |
| | | | | 2B | *S. aureus* | 99.9% |
| $10^3$ dilution | $1.178 \times 10^2$ | $1.033 \times 10^5$ | White, opaque | 3A | *S. aureus* | 99.9% |
| | | | | 3B | *S. aureus* | 99.9% |
| $10^4$ dilution | $1.178 \times 10^1$ | $7.9789 \times 10^4$ | White, opaque | 4A | No identification | n/a |
| | | | | 4B | *S. aureus* | 99.9% |
| $10^5$ dilution | 1.178 | $5.4334 \times 10^4$ | White, opaque | 5A | *S. aureus* | 99.9% |
| | | | | 5B | *S. aureus* | 99.9% |
| $10^6$ dilution | $1.178 \times 10^{-1}$ | $2.6165 \times 10^4$ | White, opaque | 6A | *S. aureus* | 99.9% |
| | | | | 6B | *S. aureus* | 99.9% |
| $10^7$ dilution | $1.178 \times 10^{-2}$ | $7.889 \times 10^3$ | White, opaque | 7A | No identification | n/a |
| | | | | 7B | *S. aureus* | 96.1% |
| $10^8$ dilution | $1.178 \times 10^{-3}$ | $3.1665 \times 10^2$ | No visible pellet | 8A | No identification | n/a |
| | | | | 8B | No identification | n/a |
| $10^9$ dilution | $1.178 \times 10^{-4}$ | n/a | No visible pellet | 9A | No identification | n/a |
| | | | | 9B | No identification | n/a |
| $10^{10}$ dilution | $1.178 \times 10^{-5}$ | n/a | No visible pellet | 10A | No identification | n/a |
| | | | | 10B | No identification | n/a |
| Negative control—matrix only | | | | | No identification | n/a |
| Negative control–vitreous only | | | | | No identification | n/a |
| Positive control—*Enterobacter aerogenes* | | | | | *Enterobacter aerogenes* | 99.9% |
| Positive control—*S. aureus* | | | | | *S. aureus* | 99.9% |

MALDI-TOF MS = Matrix-Assisted Laser Desorption/Ionization Time-of-Flight Mass Spectrometry; *S. aureus* = *Staphylococcus aureus*; n/a = not applicable.

**Table 2. Results from direct automated AST of *in vitro* endophthalmitis samples with VITEK 2.** Concentrations of *S. aureus* in vitreous humor pre-and post-incubation, McFarland units, time needed for analysis, antimicrobial agents tested, minimum inhibitory concentration, and results from VITEK 2 analysis with % identity of results compared to AST profile of positive control are depicted. Positive control was a colony of *S. aureus* analyzed per standard protocol.

| Sample | Positive control *S. aureus* | Pellet 1 | Pellet 2 | Pellet 3 | Pellet 4 |
|---|---|---|---|---|---|
| Pre-incubation (cfu/μl) | | $1.67 \times 10^5$ cfu/ul | $1.67 \times 10^4$ cfu/ul | $1.67 \times 10^3$ cfu/ul | $1.67 \times 10^2$ cfu/ul |
| Post-incubation (cfu/μl) | | $2.43 \times 10^5$ cfu/ul | $2.65 \times 10^5$ cfu/ul | $1.13 \times 10^5$ cfu/ul | $5.33 \times 10^5$ cfu/ul |
| McFarland units | 0.56 McFarland | 0.62 McFarland | 0.57 McFarland | 0.63 McFarland | 0.59 McFarland |
| Time for analysis | 8h | 9.25h | 8.75h | 8.25h | 8.75h |
| Antimicrobial agent (MIC) | | | | | |
| Beta-lactamase | Positive | Positive | Positive | Positive | Positive |
| Cefoxitin screen | Positive | Positive | Positive | Positive | Positive |
| Oxacillin | Resistant ($\geq 4$ μg/mL) | Resistant ($\geq 4$ μg/mL) | Resistant ($\geq 4$ μg/mL) | Resistant ($\geq 4$ μg/mL) | Resistant ($\geq 4$ μg/mL) |
| Cefazolin | Resistant | Resistant | Resistant | Resistant | Resistant |
| Gentamicin | Sensitive ($\leq 0.5$ μg/mL) | Sensitive ($\leq 0.5$ μg/mL) | Sensitive ($\leq 0.5$ μg/mL) | Sensitive ($\leq 0.5$ μg/mL) | Sensitive ($\leq 0.5$ μg/mL) |
| Ciprofloxacin | Intermediate (= 2 μg/mL) | Sensitive (= 1 μg/mL)* | Sensitive (= 1 μg/mL)* | Intermediate (= 2 μg/mL) | Sensitive (= 1 μg/mL)* |
| Levofloxacin | Sensitive (= 0.5 μg/mL) | Sensitive (= 0.5 μg/mL) | Sensitive (= 0.5 μg/mL) | Sensitive (= 0.5 μg/mL) | Sensitive (= 0.5 μg/mL) |
| Moxifloxacin | Sensitive ($\leq 0.25$ μg/mL) | Sensitive ($\leq 0.25$ μg/mL) | Sensitive ($\leq 0.25$ μg/mL) | Sensitive ($\leq 0.25$ μg/mL) | Sensitive ($\leq 0.25$ μg/mL) |
| Clindamycin (inducible resistance) | Negative | Negative | Negative | Negative | Negative |
| Erythromycin | Resistant ($\geq 8$ μg/mL) | Resistant ($\geq 8$ μg/mL) | Resistant ($\geq 8$ μg/mL) | Resistant ($\geq 8$ μg/mL) | Resistant ($\geq 8$ μg/mL) |
| Clindamycin | Sensitive ($\leq 0.25$ μg/mL) | Sensitive ($\leq 0.25$ μg/mL) | Sensitive ($\leq 0.25$ μg/mL) | Sensitive ($\leq 0.25$ μg/mL) | Sensitive ($\leq 0.25$ μg/mL) |
| Quinupristin/ dalfopristin | Sensitive ($\leq 0.25$ μg/mL) | Sensitive ($\leq 0.25$ μg/mL) | Sensitive ($\leq 0.25$ μg/mL) | Sensitive ($\leq 0.25$ μg/mL) | Sensitive ($\leq 0.25$ μg/mL) |
| Linezolid | Sensitive (= 2 μg/mL) | Sensitive (= 1 μg/mL)* | Sensitive (= 1 μg/mL)* | Sensitive (= 1 μg/mL)* | Sensitive (= 1 μg/mL)* |
| Vancomycin | Sensitive (= 1 μg/mL) | Sensitive (= 1 μg/mL) | Sensitive (= 1 μg/mL) | Sensitive (= 1 μg/mL) | Sensitive (= 1 μg/mL) |
| Tetracycline | Sensitive ($\leq 1$ μg/mL) | Sensitive ($\leq 1$ μg/mL) | Sensitive ($\leq 1$ μg/mL) | Sensitive ($\leq 1$ μg/mL) | Sensitive ($\leq 1$ μg/mL) |
| Tigecycline | Sensitive ($\leq 0.12$ μg/mL) | Sensitive ($\leq 0.12$ μg/mL) | Sensitive ($\leq 0.12$ μg/mL) | Sensitive ($\leq 0.12$ μg/mL) | Sensitive ($\leq 0.12$ μg/mL) |
| Nitrofurantoin | Sensitive ($\leq 16$ μg/mL) | Sensitive ($\leq 16$ μg/mL) | Sensitive ($\leq 16$ μg/mL) | Sensitive ($\leq 16$ μg/mL) | Sensitive ($\leq 16$ μg/mL) |
| Rifampicin | Sensitive ($\leq 0.5$ μg/mL) | Sensitive ($\leq 0.5$ μg/mL) | Sensitive ($\leq 0.5$ μg/mL) | Sensitive ($\leq 0.5$ μg/mL) | Sensitive ($\leq 0.5$ μg/mL) |
| Trimethoprim/ sulfamethoxazole | Sensitive ($\leq 10$ μg/mL) | Sensitive ($\leq 10$ μg/mL) | Sensitive ($\leq 10$ μg/mL) | Sensitive ($\leq 10$ μg/mL) | Sensitive ($\leq 10$ μg/mL) |
| % Identity to Positive control | | 88.9% | 88.9% | 94.4% | 88.9% |
| Sample | Positive control *S. aureus* | Pellet 5 | Pellet 6 | Pellet 7 | |
| Pre-incubation (cfu/μl) | | $1.67 \times 10^1$ cfu/ul | 1.67 cfu/ul | $1.67 \times 10^{-1}$ cfu/ul | |
| Post-incubation (cfu/μl) | | $4.98 \times 10^5$ cfu/ul | $2.40 \times 10^5$ cfu/ul | $7.00 \times 10^4$ cfu/ul | |
| McFarland units | 0.56 McFarland | 0.47 McFarland | 0.58 McFarland | 0.52 McFarland | |
| Time for analysis | 8h | 8.5h | 8.25h | 8.25h | |
| Antimicrobial agent (MIC) | | | | | |
| Beta-lactamase | Positive | Positive | Positive | Positive | |
| Cefoxitin screen | Positive | Positive | Positive | Positive | |
| Oxacillin | Resistant ($\geq 4$ μg/mL) | Resistant ($\geq 4$ μg/mL) | Resistant ($\geq 4$ μg/mL) | Resistant ($\geq 4$ μg/mL) | |
| Cefazolin | Resistant | Resistant | Resistant | Resistant | |

*(Continued)*

**Table 2.** (Continued)

| | | | | |
|---|---|---|---|---|
| Gentamicin | Sensitive ($\leq$ 0.5 µg/mL) | Sensitive ($\leq$ 0.5 µg/mL) | Sensitive ($\leq$ 0.5 µg/mL) | Sensitive ($\leq$ 0.5 µg/mL) |
| Ciprofloxacin | Intermediate (= 2 µg/mL) | Sensitive (= 1 µg/mL)* | Sensitive (= 1 µg/mL)* | Sensitive (= 1 µg/mL)* |
| Levofloxacin | Sensitive (= 0.5 µg/mL) | Sensitive (= 0.5 µg/mL) | Sensitive (= 0.5 µg/mL) | Sensitive (= 0.5 µg/mL) |
| Moxifloxacin | Sensitive ($\leq$ 0.25 µg/mL) | Sensitive ($\leq$ 0.25 µg/mL) | Sensitive ($\leq$ 0.25 µg/mL) | Sensitive ($\leq$ 0.25 µg/mL) |
| Clindamycin (inducible resistance) | Negative | Negative | Negative | Negative |
| Erythromycin | Resistant ($\geq$ 8 µg/mL) | Resistant ($\geq$ 8 µg/mL) | Resistant ($\geq$ 8 µg/mL) | Resistant ($\geq$ 8 µg/mL) |
| Clindamycin | Sensitive ($\leq$ 0.25 µg/mL) | Sensitive ($\leq$ 0.25 µg/mL) | Sensitive ($\leq$ 0.25 µg/mL) | Sensitive ($\leq$ 0.25 µg/mL) |
| Quinupristin/dalfopristin | Sensitive ($\leq$ 0.25 µg/mL) | Sensitive ($\leq$ 0.25 µg/mL) | Sensitive ($\leq$ 0.25 µg/mL) | Sensitive ($\leq$ 0.25 µg/mL) |
| Linezolid | Sensitive (= 2 µg/mL) | Sensitive (= 2 µg/mL) | Sensitive (= 2 µg/mL) | Sensitive (= 1 µg/mL)* |
| Vancomycin | Sensitive (= 1 µg/mL) | Sensitive (= 1 µg/mL) | Sensitive (= 1 µg/mL) | Sensitive (= 1 µg/mL) |
| Tetracycline | Sensitive ($\leq$ 1 µg/mL) | Sensitive ($\leq$ 1 µg/mL) | Sensitive ($\leq$ 1 µg/mL) | Sensitive ($\leq$ 1 µg/mL) |
| Tigecycline | Sensitive ($\leq$ 0.12 µg/mL) | Sensitive ($\leq$ 0.12 µg/mL) | Sensitive ($\leq$ 0.12 µg/mL) | Sensitive ($\leq$ 0.12 µg/mL) |
| Nitrofurantoin | Sensitive ($\leq$ 16 µg/mL) | Sensitive ($\leq$ 16 µg/mL) | Sensitive ($\leq$ 16 µg/mL) | Sensitive ($\leq$ 16 µg/mL) |
| Rifampicin | Sensitive ($\leq$ 0.5 µg/mL) | Sensitive ($\leq$ 0.5 µg/mL) | Sensitive ($\leq$ 0.5 µg/mL) | Sensitive ($\leq$ 0.5 µg/mL) |
| Trimethoprim/sulfamethoxazole | Sensitive ($\leq$ 10µg/mL) | Sensitive ($\leq$ 10µg/mL) | Sensitive ($\leq$ 10µg/mL) | Sensitive ($\leq$ 10µg/mL) |
| % Identity to Positive control | | 94.4% | 94.4% | 88.9% |

Asterisks (*) indicate AST results that deviated from the AST results of the positive control. AST = antimicrobial susceptibility test; *S. aureus* = *Staphylococcus aureus*.

## MALDI-TOF MS analysis

Sterile inoculating loops were used to apply portions of the bacterial pellet onto a spot on the target plate (Fig 2). One bacterial pellet was applied to two spots. Following the *in vitro* endophthalmitis model, the VH that was inoculated with more dilute concentrations of BS ($\leq 1.178 \times 10^{-3}$ cfu/µl) did not produce visible bacterial pellets, and so 480 µl of the supernatant was aspirated from the top of the sample and the remaining 100 µl were mixed with pipetting. If there were no visible pellets to smear onto the target plate, 1µl of fluid was applied onto 2 spots. Positive controls of *S. aureus* and *Enterobacter aurogenes*, and negative controls of matrix solution (α-cyano-4-hydroxycinnamic acid) and sterile VH were applied to one spot each. Each spot was overlaid with 1µl of matrix per manufacturer's protocol and allowed to dry sufficiently. The target plate was subsequently inserted into the MALDI-TOF MS machine.

## VITEK 2 analysis for AST

After bacterial pellets were produced with the *in vitro* endophthalmitis model, the pellets were resuspended in 2ml of 0.45% sterile saline to reach between 0.5 and 0.63 McFarland units per the manufacturer's instructions. The VH that was inoculated with more dilute concentrations of VBS ($\leq 1.67 \times 10^{-2}$ cfu/µl) did not produce visible pellets and the minimum 0.5 McFarland units was not achievable for VITEK 2 analysis, and so those samples were excluded from analysis. Seven pellets were analyzed with VITEK 2. A colony of *S. aureus* grown on TSA agar was picked with a sterile inoculating loop and resuspended in 2ml of 0.45% sterile saline and

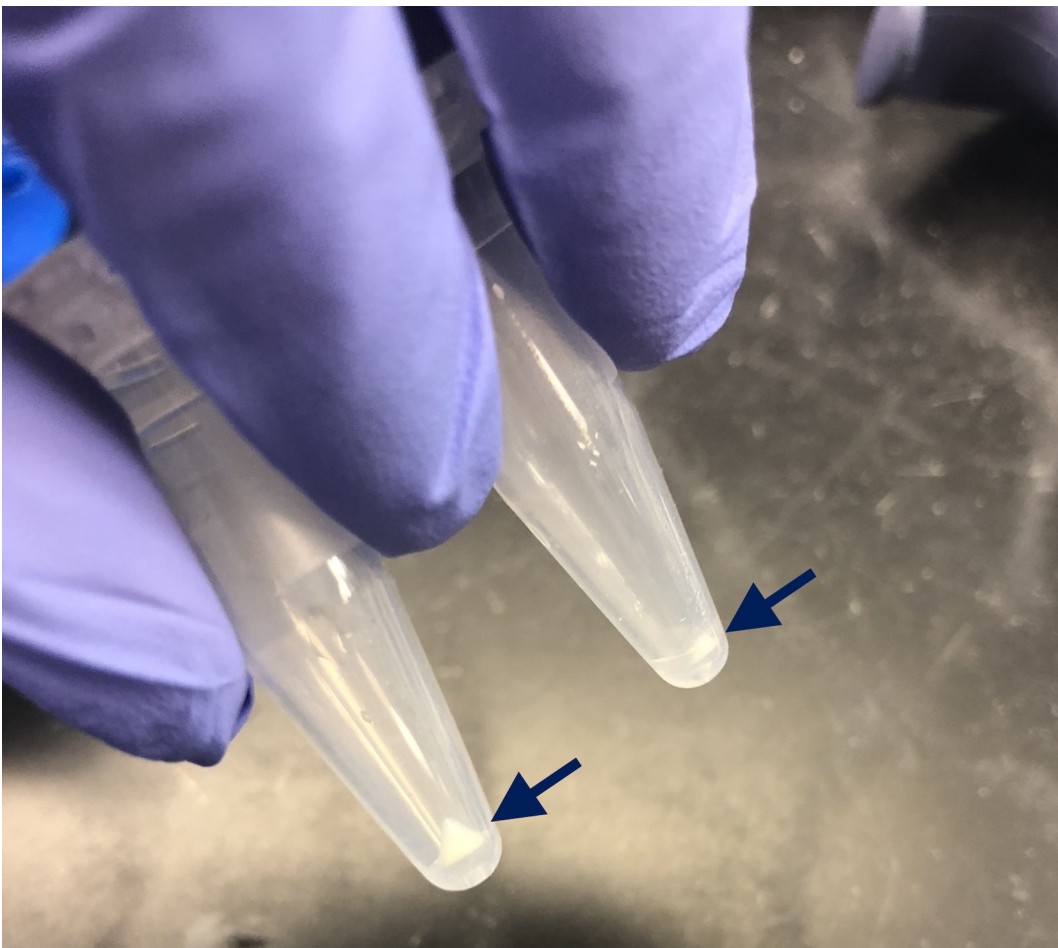

**Fig 1. Bacterial pellets formed following centrifugation of vitreous humor samples from *in vitro* endophthalmitis.**
Bacterial pellets (indicated by arrows) were white with mucoid consistency, and progressively became smaller with each subsequent dilution. Below are pellets formed after incubation of VH with $10^6$ dilution of bacterial stock (left tube) and $10^7$ dilution of bacterial stock (right tube). VH = vitreous humor.

analyzed as a positive control per standard protocol. The samples were identified as *S. aureus* in the program per manufacturer's instructions and were loaded into the appropriate cartridge (VITEK 2 AST-GP67 cartridge, bioMérieux) (Fig 3). The susceptibilities to 18 antimicrobial agents were profiled with results of "Sensitive," "Intermediate," and "Resistant" (Table 2). Minimum inhibitory concentration (MIC) needed for each agent was also determined. Reactions to beta-lactamase agents and cefotoxin screen were determined as "Positive" or "Negative."

## Results

### Pathogen identification from *in vitro* endophthalmitis sample by MALDI-TOF MS

The VH that were inoculated with at least $1.178 \times 10^{-2}$ cfu/µl of bacteria produced visible pellets, corresponding to the VH inoculated with at least $10^7$ dilution of the bacterial stock (Table 1). The pellets were white and opaque with mucoid consistency, and the size of the pellets gradually decreased with decreasing concentration of bacteria (Fig 1). The VH inoculated with

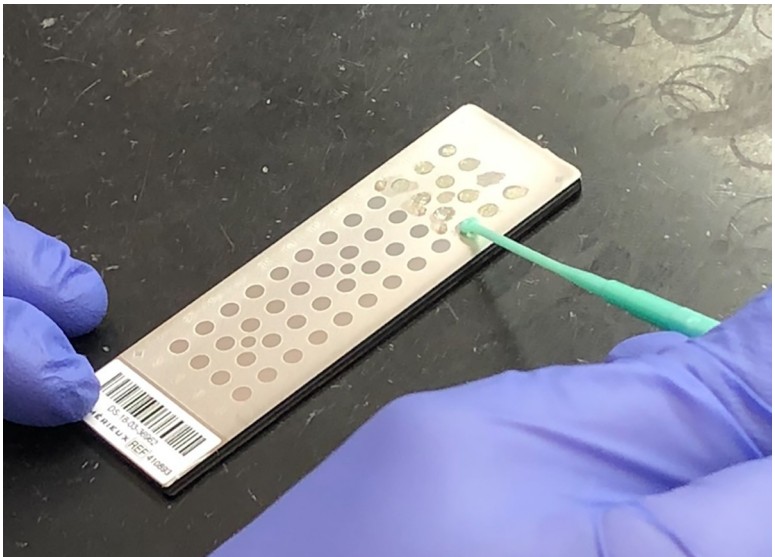

**Fig 2. Applying portions of pellets onto spots of the MALDI-TOF MS target plate with a sterile inoculating loop.**

bacterial stock dilutions of $10^8$–$10^{10}$ ($\leq 1.178 \times 10^{-3}$ cfu/μl) produced no visible pellets following centrifugation (Table 1).

Upon analysis by MALDI-TOF MS, all samples that produced visible pellets had positive identification of *S. aureus* with confidence values of $\geq 95\%$. The minimum concentration of bacteria in VH that resulted in a positive and accurate identification following incubation was

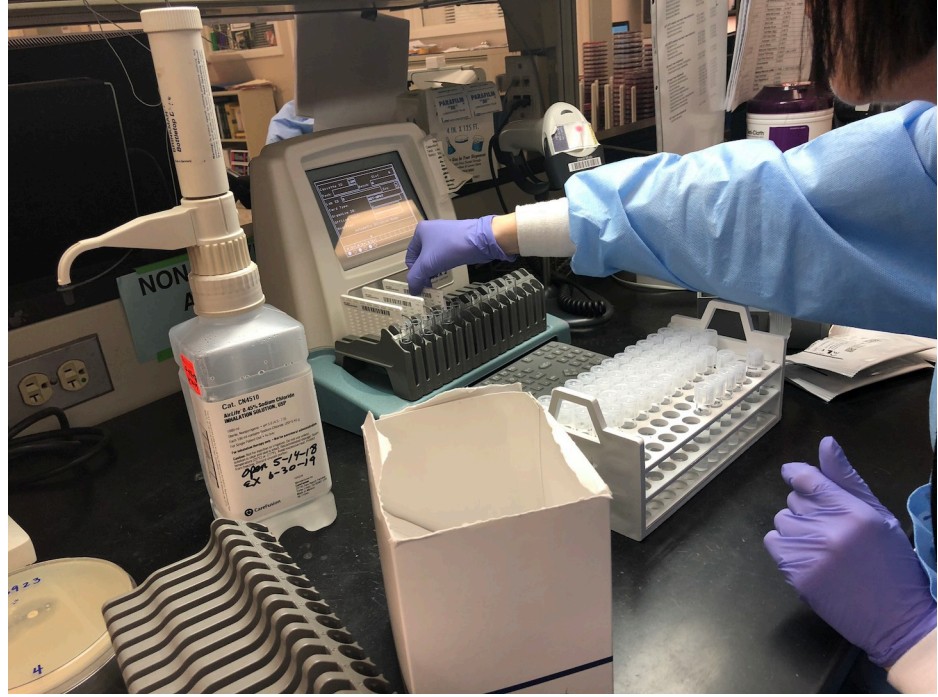

**Fig 3. Inserting AST GP67 cartridges (VITEK 2) into tubes containing pellets resuspended in 0.45% sterile saline.**

7.889x10$^3$ cfu/μl of VH, which had a confidence value of 96.1%. The samples that had lower concentrations of bacteria gave no identification from MALDI-TOF MS analysis.

One out of two spots from a pellet formed from VH with 7.98x10$^4$ cfu/μl (VH inoculated with 1.178x10$^1$ cfu/μl), and one spot from a pellet formed from VH with 7.89x10$^3$ cfu/μl (VH inoculated with 1.178x10$^{-2}$ cfu/μl), gave no identification from analysis, likely due to inadequate application of the pellet onto the target plate. The other spots from each of the samples gave a single, positive identification of *S. aureus*. No other organisms were identified from the VH.

Positive controls of a colony of *S. aureus* and a colony of *Enterobacter aerogenes* were identified with 99.9% confidence value. Negative controls of matrix and uninoculated VH showed no identification of organisms, indicating there was no contamination of the experimental materials used.

The time for sample processing was 20 minutes. The time for applying a single spot on the target plate was 30 seconds. The time for analysis of a single spot on the target plate was less than 60 seconds by MALDI-TOF MS. The results were available less than 5 minutes following analysis. Overall, the time for pathogen identification with MALDI-TOF MS, from sample processing to the acquisition of results, was less than 30 minutes for a single sample.

### Pathogen antimicrobial susceptibility by VITEK 2

The first seven pellets were analyzed with VITEK 2 because they reached adequate McFarland units for VITEK 2 analysis following resuspension in 0.45% sterile saline (Table 2).

Three samples (pellets 3, 5, 6) from the in vitro endophthalmitis model gave 17/18, or 94.4%, identity to the AST profile of the positive control of *S. aureus*. Pellets 5 and 6 showed sensitivity to ciprofloxacin at a MIC of 1 μg/mL, compared to the positive control's intermediate reaction to ciprofloxacin at a MIC of 2 μg/mL. Pellet 3 showed sensitivity to linezolid at a MIC of 1 μg/mL, compared to the positive control's sensitivity at a MIC of 2 μg/mL.

Four samples (pellets 1, 2, 4, 7) gave 16/18, or 88.9%, identity to the positive control *S. aureus*. All these pellets differed from the positive control AST profile in their sensitivity to ciprofloxacin at 1μg/mL compared to positive control's intermediate reaction at a MIC of 2 μg/mL, and sensitivity to linezolid at 1 μg/mL compared to the positive control's sensitivity at a MIC of 2 μg/mL.

One sample (pellet 5) was diluted to a McFarland of 0.47, less than the minimal threshold of 0.5 McFarland, but gave an AST profile with 94.44% identity to the positive control.

The range of time for antimicrobial susceptibility testing of the pellets was 8–9.25 hours, with an average of 8.5 hours.

### Discussion

The role of MALDI-TOF MS and automated AST may mark a shift in microbiological methodology, at a time where the value of targeted therapy prevails in the face of increasing pathogen resistance and its consequent mortality and costs to healthcare.[9] Physicians treating patients with endophthalmitis must be aware of the potentially fatal consequences to vision, and the prognosis for patients can be very poor. In one study, 21.7% of eyes were reported of never being able to regain their baseline visual acuity after 6 months, and in another study up to 10% of eyes were reported to suffer from complete vision loss[1,21,22].

Using our *in vitro* endophthalmitis model, we demonstrate that the minimal processing (centrifugation of samples and washings with sterile water) of VH followed by direct analysis of the VH with MALDI-TOF MS without prior culture could lead to the identification of the pathogen with confidence values of up to 99.9% in most cases. A score of ≥60% indicates

species-level identification[10–13]. We were also able to demonstrate that the minimum concentration of bacteria that led to identification of the pathogen was $7.889 \times 10^6$ cfu/μl. It has previously been established that positive identification of a microorganism requires at least $10^4$ cells in a sample analyzed with MALDI-TOF MS[14]. We found that the time required, from sample processing to pathogen identification by MALDI-TOF MS, was less than 30 minutes for each sample. A previous study showed that the use of MALDI-TOF MS could decrease the time required for positive pathogen identification in human cases of endophthalmitis by up to 109 hours [23].

Compared to traditional identification methods, MALDI-TOF MS holds high potential as an analytical tool for the characterization of different types of microorganisms, and has a gain of time of days[13,15,23]. From the samples that formed pellets, only two out of the fourteen spots gave no identification likely due to inadequate application of the pellets onto the target plate. Every pellet analyzed gave a positive single identification of *S. aureus*. The samples that produced no visible pellets (VH inoculated with $10^8$–$10^{10}$ dilutions of *S. aureus*) gave no identification, likely due to insufficient number of microorganisms present in the sample for MALDI-TOF MS analysis.

We also demonstrated that an automated AST system, VITEK 2, can be used to directly analyze the VH of our *in vitro* model of endophthalmitis without prior culture and determine the AST of the pathogen with up to 94.44% accuracy compared to the positive control *S. aureus*. Our findings show that the growth of the causative pathogen through standard culturing methods is not necessary for analysis with VITEK 2, given that the minimum turbidity (0.5 McFarland units) of the analyzed material is met. The ability to directly determine the AST of a pathogen without growing the organism in culture, could significantly reduce the time and resources, and optimize the treatment strategy of patients with endophthalmitis. In our experiments, the range of time needed for the attainment of AST profiles with VITEK 2 was 8–9.25 hours, in contrast to the multiple days needed to successfully grow pathogens via culture with the current, conventional methodology of attaining AST results [20,23].

Although endophthalmitis is a rare condition, its incidence is likely to rise with the anticipated rise of ocular procedures. Cataract surgery and intravitreal injections are among the most commonly performed procedures in ophthalmology and medicine in general, and each procedure involves a risk for infection[9]. As the general population ages, the incidence of cataract surgeries is projected to increase dramatically in developed and developing countries, and the advent of new intravitreal agents for a broad array of retinal diseases, including neovascular age-related macular degeneration and diabetic macular edema, is likely to lead to an increase in intraocular injections performed on a daily basis[22,24–26]. Guidelines for preoperative preparation and sterile procedural techniques have likely aided in keeping reported rates of endophthalmitis low following cataract surgery (0.012 to 1.3%) and intravitreal injections (0.016 to 0.2%)[1]. However, the frequency with which these procedures are performed makes the risk of infectious complications a point of serious concern for patient care in ophthalmology.

During the time it takes to isolate an organism, the administration of broad-spectrum antibiotics typically is initiated to salvage the eye. Current recommendations for intravitreal antibiotics include vancomycin (1mg/0.1mL) and ceftazidime (2.25mg/0.1mL). However, broad-spectrum antibiotics can complicate the course of management because they can put the patient at future risk of succumbing to infection from an antimicrobial-resistant organism. Recent reports of endophthalmitis caused by vancomycin- and ceftazidime-resistant organisms underscore the importance of carefully considering the use of these agents for treatment [11,19,27,28]. Poor visual outcomes have especially been noted with Gram-negative organisms and certain Gram-positive organisms, especially those with resistance to broad spectrum

antibiotics[9,19,29]. There is also a risk for retinal detachment from endophthalmitis, especially in cases of infection by more virulent pathogens and higher severity of disease at presentation[30,31]. The administration of antibiotics at the time of patient presentation may also prevent the identification of organisms through traditional microbiological methods.

Successful pathogen identification has been demonstrated with the direct application of cerebrospinal fluid and urine from cases of meningitis and urinary tract infections, respectively, onto MALDI-TOF MS without prior culture[14,32]. A case of pathogen identification through the direct application of MALDI-TOF MS on a vitreous sample from endophthalmitis, without prior culture, has recently been reported as well [33]. A wider breadth of organisms including other bacteria, mycobacteria, fungi, and polymicrobial infections could also be investigated with MALDI-TOF MS. Studies with *in vivo* animal models and human samples of endophthalmitis are needed to validate the clinical value of MALDI-TOF MS and automated AST in endophthalmitis. The potential effects of inflammatory responses on the specificity and sensitivity of direct analysis of samples with MALDI-TOF MS also require investigation.

There are limitations to the use of MALDI-TOF MS however, as the scope of pathogen identification is limited by the breadth of organisms established in the database of the specific biotyper software that is employed. For clinical applicability of the techniques we describe, there must be an adequate quantity of bacteria present in intraocular samples obtained from patients as well. Studies should also determine the ability of pathogen identification following antibiotic administration because unlike conventional microbiological methods, which require that the organism be intact or alive for proper identification, MALDI-TOF MS only requires the presence of particles of the culprit organism.

Directly applying endophthalmitis vitreous samples onto MALDI-TOF MS and VITEK 2 presents a promising new technique for the rapid identification of pathogens in the setting of endophthalmitis. Our findings demonstrate the proof-of-concept that the direct analysis of intraocular samples with these techniques could be used as an improved supplemental method to provide the rapid accuracy needed for proper treatment strategy in endophthalmitis. Further studies are needed to further validate potential clinical applications.

## Acknowledgments

We would like to extend our thanks to the Illinois Society for the Prevention of Blindness (ISPB), the members of the Missiakas-Schneewind laboratory at the University of Chicago, and the University of Chicago Medical Center Clinical Microbiology Laboratory for generously allowing us access to resources necessary to pursue our project.

## Author Contributions

**Conceptualization:** Dominique Missiakas, Olaf Schneewind, Kathleen G. Beavis, Dimitra Skondra.

**Data curation:** Lindsay Y. Chun, Dimitra Skondra.

**Formal analysis:** Lindsay Y. Chun, Dimitra Skondra.

**Funding acquisition:** Lindsay Y. Chun, Kathleen G. Beavis, Dimitra Skondra.

**Investigation:** Lindsay Y. Chun, Rose C. Dimitroyannis, Dominique Missiakas, Olaf Schneewind, Kathleen G. Beavis, Dimitra Skondra.

**Methodology:** Lindsay Y. Chun, Laura Dolle-Molle, Cindy Bethel, Dominique Missiakas, Olaf Schneewind, Kathleen G. Beavis, Dimitra Skondra.

**Project administration:** Lindsay Y. Chun, Cindy Bethel, Dominique Missiakas, Olaf Schneewind, Kathleen G. Beavis, Dimitra Skondra.

**Resources:** Lindsay Y. Chun, Laura Dolle-Molle, Cindy Bethel, Dominique Missiakas, Olaf Schneewind, Kathleen G. Beavis, Dimitra Skondra.

**Software:** Lindsay Y. Chun, Laura Dolle-Molle, Cindy Bethel, Dominique Missiakas, Olaf Schneewind, Kathleen G. Beavis, Dimitra Skondra.

**Supervision:** Lindsay Y. Chun, Cindy Bethel, Dominique Missiakas, Olaf Schneewind, Kathleen G. Beavis, Dimitra Skondra.

**Validation:** Lindsay Y. Chun, Dominique Missiakas, Olaf Schneewind, Kathleen G. Beavis, Dimitra Skondra.

**Visualization:** Lindsay Y. Chun, Dominique Missiakas, Olaf Schneewind, Kathleen G. Beavis, Dimitra Skondra.

**Writing – original draft:** Lindsay Y. Chun, Dominique Missiakas, Olaf Schneewind, Kathleen G. Beavis, Dimitra Skondra.

**Writing – review & editing:** Lindsay Y. Chun, Blake L. Williams, Sidney A. Schechet, Seenu M. Hariprasad, Dominique Missiakas, Olaf Schneewind, Kathleen G. Beavis, Dimitra Skondra.

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
