## [Decision Letter · Decision Letter 0]

21 Oct 2019

PONE-D-19-26072

Rapid Pathogen Identification and Antimicrobial Susceptibility Testing in in vitro Endophthalmitis with Matrix Assisted Laser Desorption-Ionization Time-of-Flight Mass Spectrometry and VITEK 2 without Prior Culture

PLOS ONE

Dear Dr. Skondra,

Thank you for submitting your manuscript to PLOS ONE. After careful consideration, we feel that it has merit but does not fully meet PLOS ONE’s publication criteria as it currently stands. Therefore, we invite you to submit a revised version of the manuscript that addresses the points raised during the review process.

This is an exciting new study and idea. both reviewers find it so as well. There are some minor issues that can easily be addressed and we would welcome the revised version.

We would appreciate receiving your revised manuscript by Dec 05 2019 11:59PM. To enhance the reproducibility of your results, we recommend that if applicable you deposit your laboratory protocols in protocols.io, where a protocol can be assigned its own identifier (DOI) such that it can be cited independently in the future. For instructions see: http://journals.plos.org/plosone/s/submission-guidelines#loc-laboratory-protocols

We look forward to receiving your revised manuscript.

Kind regards,

Demetrios G. Vavvas

Academic Editor

PLOS ONE

**Journal Requirements:**

**Comments to the Author**

1. Is the manuscript technically sound, and do the data support the conclusions?

Reviewer #1: Yes

Reviewer #2: Yes

2. Has the statistical analysis been performed appropriately and rigorously? 

Reviewer #1: Yes

Reviewer #2: Yes

3. Have the authors made all data underlying the findings in their manuscript fully available?

Reviewer #1: Yes

Reviewer #2: Yes

4. Is the manuscript presented in an intelligible fashion and written in standard English?

Reviewer #1: Yes

Reviewer #2: Yes

5. Review Comments to the Author

Reviewer #1: Thank you for submitting your research work to PLOS One.

I find this article very interesting. Congratulations on the excellent work of the entire team and on the well-written manuscript. Despite its limitations, I agree that this technology can become a powerful tool for rapid identification of pathogens in patients with endophthalmitis.

There is a typo: p.6 the word Design (currently is DEISGN).

Thank you.

Reviewer #2: This is a very interesting study but would recommend the following modifications:

1. Please modify the last sentence of the first paragraph of Introduction, which states that 23-35% of eyes affected by endophthalmitis are left with no light perception. This very high rate of NLP visual outcome may be true for bleb-related endophthalmitis (the reference the authors give), but this is not true for other common types of endophthalmitis such as post-cataract endophthalmitis. Visual outcome is highly associated with pathogen; in coagulase-negative endophthalmitis - the major pathogen of post-cataract endophthalmitis -- NLP vision is very rare (<4%).

2. Please add a discussion of this study versus a similar one (not listed in Reference list) published in 2017 by Song Z, et al. "Using MALDI-TOF MS to test Staphylococcus aureus-infected vitreous". Mol Vis. 2017; 23: 407–415. The Song study used porcine eyes injected with S. aureus and directly tested the resulting infected vitreous with MALDI-TOF, so is similar in some regards to the present study -- results may in fact be complimentary. Also, the current study mentions a MALDI-TOF result in a patient with culture-negative endophthalmitis (ref 33), and the Song 2017 study mentions results from 2 patients with culture-positive endophthalmitis in which MALDI-TOF and standard culture yielded the same organism identification. This also may be worth noting.

6. PLOS authors have the option to publish the peer review history of their article (what does this mean?). If published, this will include your full peer review and any attached files.

Reviewer #1: No

Reviewer #2: No

---

## [Author Response · Author response to Decision Letter 0]

29 Nov 2019

Journal Requirements:

Comments to the Author

1. Is the manuscript technically sound, and do the data support the conclusions?

Reviewer #1: Yes

Reviewer #2: Yes

2. Has the statistical analysis been performed appropriately and rigorously?

Reviewer #1: Yes

Reviewer #2: Yes

3. Have the authors made all data underlying the findings in their manuscript fully available?

Reviewer #1: Yes

Reviewer #2: Yes

4. Is the manuscript presented in an intelligible fashion and written in standard English?

Reviewer #1: Yes

Reviewer #2: Yes

5. Review Comments to the Author

Reviewer #1: Thank you for submitting your research work to PLOS One.

I find this article very interesting. Congratulations on the excellent work of the entire team and on the well-written manuscript. Despite its limitations, I agree that this technology can become a powerful tool for rapid identification of pathogens in patients with endophthalmitis.

There is a typo: p.6 the word Design (currently is DEISGN).

Thank you; we have edited the typo.

Reviewer #2: This is a very interesting study but would recommend the following modifications:

1. Please modify the last sentence of the first paragraph of Introduction, which states that 23-35% of eyes affected by endophthalmitis are left with no light perception. This very high rate of NLP visual outcome may be true for bleb-related endophthalmitis (the reference the authors give), but this is not true for other common types of endophthalmitis such as post-cataract endophthalmitis. Visual outcome is highly associated with pathogen; in coagulase-negative endophthalmitis - the major pathogen of post-cataract endophthalmitis -- NLP vision is very rare (<4%).

Thank you; this sentence has been modified to state:

“Depending on the causative pathogen and pathophysiology, infectious endophthalmitis can lead to poor visual outcomes in the affected eye; for example, the incidence of no light perception ranges from 23% to 35% in bleb-related endophthalmitis(3,4).”

2. Please add a discussion of this study versus a similar one (not listed in Reference list) published in 2017 by Song Z, et al. "Using MALDI-TOF MS to test Staphylococcus aureus-infected vitreous". Mol Vis. 2017; 23: 407–415. The Song study used porcine eyes injected with S. aureus and directly tested the resulting infected vitreous with MALDI-TOF, so is similar in some regards to the present study -- results may in fact be complimentary. Also, the current study mentions a MALDI-TOF result in a patient with culture-negative endophthalmitis (ref 33), and the Song 2017 study mentions results from 2 patients with culture-positive endophthalmitis in which MALDI-TOF and standard culture yielded the same organism identification. This also may be worth noting.

The discussion has been modified to state:

“Another group previously reported on the implementation of MALDI-TOF MS in identifying a strain of S. aureus from in-vitro and ex-vivo models of endophthalmitis with porcine eyes, and in identifying the causative organisms in culture-positive human endophthalmitis samples, (34). However, we aimed to anticipate the extrapolation of our methods to potential clinical scenarios, and we thus optimized our methodology to require relatively minimal steps in sample preparation, clearly delineate the parameters and materials we employed, and help establish a range of minimum time needed for organism identification with MALDI-TOF MS.

Although the previously reported findings on the direct analysis of samples with MALDI-TOF MS support our study, a wider breadth of organisms including other bacteria, mycobacteria, fungi, and polymicrobial infections should also be investigated with MALDI-TOF MS.” 

6. PLOS authors have the option to publish the peer review history of their article (what does this mean?). If published, this will include your full peer review and any attached files.

Do you want your identity to be public for this peer review? For information about this choice, including consent withdrawal, please see our Privacy Policy.

Reviewer #1: No

Reviewer #2: No

---

## [Decision Letter · Decision Letter 1]

12 Dec 2019

Rapid Pathogen Identification and Antimicrobial Susceptibility Testing in in vitro Endophthalmitis with Matrix Assisted Laser Desorption-Ionization Time-of-Flight Mass Spectrometry and VITEK 2 without Prior Culture

PONE-D-19-26072R1

Dear Dr. Skondra,

We are pleased to inform you that your manuscript has been judged scientifically suitable for publication and will be formally accepted for publication once it complies with all outstanding technical requirements.

With kind regards,

Demetrios G. Vavvas

Academic Editor

PLOS ONE

Additional Editor Comments (optional):

Reviewers' comments:

Reviewer's Responses to Questions

**Comments to the Author**

1. If the authors have adequately addressed your comments raised in a previous round of review and you feel that this manuscript is now acceptable for publication, you may indicate that here to bypass the “Comments to the Author” section, enter your conflict of interest statement in the “Confidential to Editor” section, and submit your "Accept" recommendation.

Reviewer #2: All comments have been addressed

2. Is the manuscript technically sound, and do the data support the conclusions?

Reviewer #2: Yes

3. Has the statistical analysis been performed appropriately and rigorously? 

Reviewer #2: I Don't Know

4. Have the authors made all data underlying the findings in their manuscript fully available?

Reviewer #2: Yes

5. Is the manuscript presented in an intelligible fashion and written in standard English?

Reviewer #2: Yes

6. Review Comments to the Author

Reviewer #2: (No Response)

7. PLOS authors have the option to publish the peer review history of their article (what does this mean?). If published, this will include your full peer review and any attached files.

Reviewer #2: No

---

## [Editor Report · Acceptance letter]

17 Dec 2019

PONE-D-19-26072R1 

Rapid Pathogen Identification and Antimicrobial Susceptibility Testing in *in vitro* Endophthalmitis with Matrix Assisted Laser Desorption-Ionization Time-of-Flight Mass Spectrometry and VITEK 2 without Prior Culture 

Dear Dr. Skondra:

I am pleased to inform you that your manuscript has been deemed suitable for publication in PLOS ONE. Congratulations! Your manuscript is now with our production department. 

With kind regards,

on behalf of

Dr. Demetrios G. Vavvas 

Academic Editor

PLOS ONE